# The European Hare (*Lepus europaeus*) as a Biomonitor of Lead (Pb) and Cadmium (Cd) Occurrence in the Agro Biotope of Vojvodina, Serbia

**DOI:** 10.3390/ani12101249

**Published:** 2022-05-12

**Authors:** Dejan Beuković, Marko Vukadinović, Saša Krstović, Miroslava Polovinski-Horvatović, Igor Jajić, Zoran Popović, Vukan Lavadinović, Miloš Beuković

**Affiliations:** 1Department of Animal Science, Faculty of Agriculture, University of Novi Sad, 21000 Novi Sad, Serbia; dejan.beukovic@stocarstvo.edu.rs (D.B.); marko.vukadinovic@stocarstvo.edu.rs (M.V.); sasa.krstovic@stocarstvo.edu.rs (S.K.); igor.jajic@stocarstvo.edu.rs (I.J.); milos.beukovic@stocarstvo.edu.rs (M.B.); 2Institute of Animal Science, Faculty of Agriculture, University of Belgrade, 11000 Belgrade, Serbia; zpopovic@agrif.bg.ac.rs; 3Faculty of Forestry, University of Belgrade, 11000 Belgrade, Serbia; vukan.lavadinovic@sfb.bg.ac.rs

**Keywords:** hare, liver, heavy metals, lead, cadmium

## Abstract

**Simple Summary:**

Heavy metals such as lead and cadmium are pollutants and can be found in different biotopes. Hares, due to their natural habitat, numbers, range and choice of food are good bioindicators of the occurrence of heavy metals. Vojvodina, northern province of Republic of Serbia, has intensive agriculture with about 77.8% of agricultural land. In such agricultural biotope, we discovered that out of the two studied heavy metals in the liver of hare’s it seems that the occurrence of lead is more of the importance. In the majority of locations the cadmium concentrations were in the permitted level by Serbian legislative. However, the situation with the occurrence of lead was completely different. Only on the two out of seventeen locations average concentrations were in the permitted level.

**Abstract:**

The aim of this study was to investigate the occurrence of two heavy metals, lead and cadmium, in European hare liver samples, collected in agro biotope of northern Serbian province Vojvodina. Heavy metals such as lead (Pb) and cadmium (Cd) do not have any biological function in the animal body; however, they can be found due to the pollution in the environment. For the purpose of this study, in 196 samples from 17 different locations hare livers were analyzed for the occurrence of lead and cadmium. All samples were taken from hares harvested during the regular hunting season. The average value for lead in all analyzed samples was 884 µg/kg fresh weight (fw), with the range 59–3700 µg/kg fw. Only samples from two locations had the average concentration of lead which was within the permitted limit by the Serbian regulation. The average cadmium level in all samples was 243 µg/kg fw. The range of all samples was from 0 to 1414 µg/kg fw. Our research indicates that out of two investigated heavy metals, the occurrence of lead is more common and at a higher concentration in the agricultural development region of Vojvodina.

## 1. Introduction

The awareness about pollutants and the sources of pollution has recently increased in the Republic of Serbia. Heavy metals such as lead (Pb) and cadmium (Cd) can be indicators of pollution. Sometimes the occurrence of heavy metals in the environment can be a result of natural activities; however, much more often they are the result of anthropological activities. The increased level of lead can often be found in the proximity of busy highways, thermo power plants, heavy industry, or in the proximity of lead smelters [1,2,3,4,5,6]. Use of lead ammunition in hunting is also considered to be a source of environmental pollution [7,8]. Today, fossil fuels used in Serbia for engines are unleaded; however, up until 2009 leaded fuel was still used. The half-life of lead in the environment, especially soil, is very long. Cadmium is a heavy metal which can be found in increased amounts in the proximity of heavy industry or mining. Cadmium and lead can be found as impurities in some chemicals, mostly fertilizers that are used in agriculture and can be reasons for the increased amounts of these heavy metals in agricultural soil [9].

Lead and cadmium do not have any biological function in the organism. The content of heavy metals that can be found in animal tissue is regulated through national legislation. In the edible tissues used for human consumption, maximum levels for cadmium and lead are 0.5 mg/kg [10]. It should be mentioned that due to the nature of animals’ metabolism, the amount of cadmium is several times higher in the kidney [11,12,13]. Therefore, the maximum level of cadmium in kidney tissue is 1 mg/kg [10].

The source of the lead can be soil, water, or air. Once absorbed, lead is deposited mostly in the liver and bones [14]. The half-life of lead is very long; in the human body it is approximately 10–30 years for lead deposited in the bones [15]. It can be a contaminant of fertilizer and by this route enters the soil, water, and, consequently, plants [9]. The deposition sites for cadmium are primarily kidneys and liver and only a small part is stored in bones. The half-life is also very long, from 10 to 30 years. Cadmium in some cases causes bone demineralization [16].

The autonomous province of Vojvodina, northern part of the Republic of Serbia is a distinctly agricultural region with its geographical regions Bačka, Banat, and Srem. The area of AP Vojvodina is 21,614 km^2^ with about 77.8% of agricultural land [17]. Intensive agriculture with predominantly crop production is practiced in this region. Moreover, Serbia is one of the largest corn exporters in Europe and among the top 10 in the world [18], with more than half of the total production being grown in Vojvodina [19].

Although Vojvodina is an agriculturally highly developed region, it is also burdened with burning environmental problems, such as the Great Bačka canal, one of the most polluted waterways in Europe or the Begej highly polluted international canal [20,21]. There is numerous data from the scattered studies on the occurrence of heavy metals in different locations and different products in the region of Vojvodina. Milk from different farms in Vojvodina in proximity of highways had increased levels of some of the investigated heavy metals [22]. During a four-year survey on the occurrence of heavy metals in the vegetables locally produced in Vojvodina, some of samples had disturbingly high levels of the investigated heavy metals such as lead, cadmium, nickel, and chromium. In more than half samples of spinach, the concentration of cadmium exceeded the maximum permissible concentration. Additionally, in 46% of samples of the same vegetable, the lead concentration also exceeded the maximum permissible concentration [23].

Wild animals are a good choice for use as biomonitors of heavy metal occurrence in some biotopes. In this regard, exclusive herbivores are a better indicator than omnivores. The variability in food and the food chain effects are sometimes the limiting factors for the use of these animals as biomonitors [24]. It is common for wild or free-living domestic animals to have an increased amount of heavy metals because they do not have controlled sources of food and water [25,26,27,28]. The European hare (*Lepus europaeus*) is small mammal, well adapted, and numerous in the agro biotope of Vojvodina, which makes it a good choice as a heavy metal biomonitor [24,29]. The home range size of the animals has an important influence on its contamination with heavy metals. For European hare, this depends on the landscape, and it is from 23 ± 76.8 ha which corresponds to approximately 0.24 km^2^ [6]. The usual route of animal exposure to lead is oral consumption. The levels of lead in hare tissues appears to be among the highest compared to some other mammals. The reasons are that the hare is a monogastric animal, with lower stomach pH compared to ruminants. Lower stomach pH results in better absorption of minerals in their diets. Hares are strictly herbivorous. The lead in the organs of hares is not only from plants. The hare is much more exposed to the soil particles which results in higher intake of soil and dust and all minerals within it [24]. The European hare (*Lepus europaeus*) is a regularly hunted game animal, harvested numerous times during the hunting season. With its number and presence throughout Vojvodina, it is almost an ideal biomonitor of pollution.

For this research, a total of 196 European hare liver samples were collected from 17 locations in Vojvodina. The hare liver samples were taken during the regular hunting season. All sampled hare livers were analyzed for lead and cadmium content. At the same time, the ages of the hares were determined using the length of the dehydrated eye lenses.

## 2. Materials and Methods

Sampling was carried out during the fall of the hunting season in 2017 in 17 sites in the regions of Bačka, Banat, and Srem in Vojvodina, Serbia (Figure 1). After hunting, samples were collected for the analyses. A total of 196 samples were collected. Eye samples were used to determine the age of the brown hares on the basis of the dehydrated mass of the eye lenses.

### 2.1. Sampling

Livers of brown hare were sampled for the analysis, as target tissue in which heavy metals accumulate over time. Before analysis, all livers were checked and samples with any indication of damage caused by shooting were excluded from the analysis. This was undertaken to avoid analyzing samples that were additionally contaminated with lead from the pellet. Samples were homogenized, using a blender for food, to a uniform pulp. The material was then packed in food-grade plastic zip bags, which were clearly marked with the number registered in the sample register. The samples were stored in the freezer at −18 °C until the analysis.

Eye samples were used to determine the age of the brown hares on the basis of the dehydrated mass of the eye lenses. Samples of eye lenses were put into the formalin solution. The formalin solution was made from 36% formalin concentrate by making a 10% solution with distilled water. We used a lens mass of 280 mg as the limit value to separate two age classes. Hares with a lens mass up to 280 mg were considered to be up to 1 year old, where animals older than 1 year had an eye lens mass above 280 mg. Accordingly, the samples were divided into two groups, animals aged up to one year and over a year old. The eye lens mass limits were previously used [30,31,32].

### 2.2. Analysis of Heavy Metals

Our previously optimized method was used for the quantification of cadmium and lead in brown hare liver samples [33]. About 1 g of sample was weighed on an analytical balance EL 204—IC (Mettler Toledo, Urdorf, Switzerland) and placed in a tube where it was digested by wet ashing technique, by adding 4 mL of the concentrated nitric and hydrochloric acid mixture (3:1, *v*/*v*). The mixture was prepared with trace analysis grade nitric (Fisher Scientific, Waltham, MA, USA) and hydrochloric acid (Carl Roth, Karlsruhe, Germany). The tube was then heated in the thermal block ReactiTherm^TM^ TS-18820 (Fisher Scientific, Waltham, MA, USA) at 120 °C for 2 h. Content of lead was determined by the electrothermal atomic absorption spectrometer, model PinAAcle 900T (Perkin Elmer, Waltham, MA, USA). The instrument was calibrated using analytical standards for lead and cadmium (Fluka Analytical, Charllote, NC, USA), both 1000 mg/L in 2% nitric acid. Cadmium working solutions ranged from 0.2 to 2.0 µg/mL, while lead calibration points were from 2.0 up to 20.0 µg/mL.

For cadmium, the limit of detection (LOD) was 2.32 µg/kg, while limit of quantification (LOQ) was 7.04 µg/kg. For lead, LOD was 4.26 µg/kg and LOQ was 12.9 µg/kg.

All results are expressed on the fresh weight (fw) basis.

### 2.3. Statistical Analysis

Statistical analysis was performed using the Statistica software version 13.5.0.17 (TIBCO Software, Palo Alto, CA, USA). Descriptive statistical data were calculated and presented as mean, maximum, and minimum by location and age. Two-way analysis of variance (ANOVA) was used to indicate significant differences in metal levels among locations and age of hares (significant values, *p* ≤ 0.05). Furthermore, one-way ANOVA and Duncan’s test were used for comparison of the data.

## 3. Results

Results on the occurrence of heavy metals (Pb, Cd) in the liver of European hare are summarized in Table 1 and Table 2. The results are presented in two age groups by location, with the summarized average value per location. The average value for lead in all analyzed samples was 838 µg/kg fw. All samples contained lead, since there were no samples below the LOQ. The range of all samples was from 59 to 3700 µg/kg fw. Only samples from the two locations (locations 2 and 4) had the average concentration which was in the permitted limit by the Serbian regulation (Serbian regulation, 2011) for meat intended for human consumption. The highest average content of lead was from the samples taken from location 7 (1803 µg/kg fw).

The results of the occurrence of cadmium in the samples of hare liver are shown in Table 2. The average cadmium concentration in all samples was 243 µg/kg fw. However, the average level of cadmium in two locations (549 and 796 µg/kg fw) was above the permitted level by the Serbian legislation. The range of all samples was 0–1414 µg/kg fw. In some locations, the concentration of cadmium was below LOQ.

The mean concentration of lead in liver samples was statistically significantly different regarding the location. On the other hand, such differences between the age groups were not significant. The effects of age and location were both statistically significant concerning cadmium.

## 4. Discussion

### 4.1. Means

The results of our research indicate rather high levels of lead in the majority of samples of hare liver. Moreover, only in 2 out of 17 locations the average concentration of lead was in the permitted level for the animal tissues intended for human consumption. The average concentration of lead in all analyzed samples was 838 µg/kg fw. Another study undertaken in Serbia on the occurrence of lead in hare liver reported lower average concentrations. The average concentration found was 0.22 mg/kg fw with the maximum value of 1.72 mg/kg [29]. A maximum value of 1.72 mg/kg fw was found in samples from the location Sombor which is in Bačka, Vojvodina. However, in this research, the majority of samples were from the locations in central Serbia and just several locations from Vojvodina. Researchers from Poland found lead in samples of hare liver in the range 0.30–3.63 mg/kg. The highest amount of lead was found in the livers of hares taken from north and northwest of Cracow, probably due to the vicinity of heavy industry. The average lead content was 1.24 ± 0.59 mg/kg [34]. The concentration of lead in our samples is lower when compared to the research in Poland; however, it is considerably higher than in other studies completed in Europe in recent years [35,36,37]. The average concentration of cadmium in all samples was 243 µg/kg fw. The maximum found concentration was 1414 µg/kg fw. In another study undertaken in Serbia on the occurrence of cadmium in hare samples, the average concentration was 0.17 mg/kg ww and the maximum concentration was 0.85 mg/kg fw [29]. In the study completed in Czech Republic showed the average concentration of cadmium of 0.562 ± 0.039 mg/kg; however, the number of samples was small [37]. In Slovakia, the average concentration of cadmium in liver of *Lepus europaeus* was 0.16 mg/kg, the number of samples was 74, and the maximum value was 1.004 mg/kg. The similar average concentration of cadmium was found in hare liver in another study undertaken in Slovakia [35]. However, the average concentration of cadmium in livers of hares was much higher in the industrially developed part of Poland, with the average concentration of 1.65 ± 1.36 mg/kg ww. Moreover, the average concentration in one of the four investigated regions (location Western Zachodni) was 2.27 ± 1.59 mg/kg ww, with the maximum level of 6.63 mg/kg ww [34]. The results in our research are higher than previously found in Serbia and in some other European countries. Nevertheless, the average concentration is considerably lower than the one found in the intensive industrial regions in Poland previously mentioned [34].

### 4.2. Locations

In our study, there was a statistically significant difference in the average concentration of lead and cadmium among locations. The concentration of lead in the majority of locations was above the permitted level, only in two locations (2 and 4) concentration was at the legislation level. The highest average level was found in location 7 and it was 1803 µg/kg fw. The lowest average concentration of cadmium was found in samples from location 3 (92 µg/kg fw). Only in two locations, the average concentration of cadmium in hare livers was above the permitted level [17]. In locations 9 and 11, the average cadmium levels were 549 and 796 µg/kg fw, respectively. Compared to the other wild animals such as wild boar or roe deer the range of hare is relatively small. In each location, there is specificity in the industrial activities, intensity of agricultural activity, used fertilizers, and the proximity of highways or thermal plants. This is the most probable reason for the statistically significant differences (*p* < 0.05) shown in Table 3 in the average concentration of lead and cadmium regarding location.

### 4.3. Age

The average value of lead in samples of hare livers for the older specimens was 841 µg/kg fw, and 838 µg/kg fw for the younger animals. However, in some locations, the younger animals contained higher lead levels (locations: 2, 5, 7, 8, 9, 10, 13, 14, 15, and 16). Sometimes, a higher concentration of lead in edible tissues can be found in the younger animals. Researchers found the explanation in higher absorption of lead in younger animals [35]. It is a similar situation with other groups of wild animals; younger animals had higher levels of lead in the liver (wild boar and roe deer). The researchers concluded that younger animals have increased needs for minerals, especially calcium whose metabolic pathways are closely related with toxicokinetics of lead [38]. However, the age of hares in our research was not statistically significant concerning the concentration of lead in hare liver. This is in agreement with some other studies [29,35]. The age of animals had a significant effect on the average concentration of cadmium in the liver. The average cadmium concentration for younger animals was 203 and 308 µg/kg for older hares. This is in agreement with previous investigations [24,35].

It seems that the occurrence of cadmium is much lower compared to the lead in the livers of hares in our study. The free-living animals do not have any restrictions concerning sources of water or food. It is not unusual for them to have increased levels of heavy metals in edible tissues [39,40]. The concentration of cadmium seems to be age dependent; higher in older hares which is in accordance with other research [35]. The region of Vojvodina is burdened with a lot of ecological problems. Some of them are connected to the one of the most polluted waterways in Europe [20] or highly polluted international canal [21]. Additionally, modern agriculture is accompanied with the intensive use of fertilizers, pesticides, and different kinds of chemicals [18,41]. Moreover, the increased level of heavy metals in the Republic of Serbia can be seen in fattening animals [42], milk [22], or food and feed [23,43,44]. Similar situations can be seen also in the neighboring countries. A survey of several heavy metals in cattle, sheep, horses, and pigs in rural Croatia indicated the increased levels for some heavy metals [25].

Kanstrup et al. (2018) [45] claim that lead ammunition has adverse effects on wildlife, their habitat, or human health, and for that reason is an issue for sustainable hunting and a threat to conservation efforts. Use of such ammunition is considered to be a source of higher lead concentration in hunted animals [46,47,48]. Although samples in this study were only from undamaged livers taken immediately after the hunts, and only some hunters used lead ammunition, there is a possibility that pellet fragments contaminated the specimens which affected the obtained findings.

## 5. Conclusions

Our research indicates that out of two investigated heavy metals, the occurrence of lead is more common and in a higher concentration in the agricultural development region of Vojvodina. There was a statistically significant difference among sample locations, while results between different ages of hares appeared not to be significantly different. The concentration of cadmium was mostly in the permitted level apart from the samples from two locations. It seems that cadmium is a less common problem in the investigated locations. Based on our findings, there is a need for the continued monitoring of heavy metals. Despite our belief that it is less likely that lead ammunition contaminated specimens, such outcome is possible which limits this research. Thus, further research is needed in that direction.

## Figures and Tables

**Figure 1 animals-12-01249-f001:**
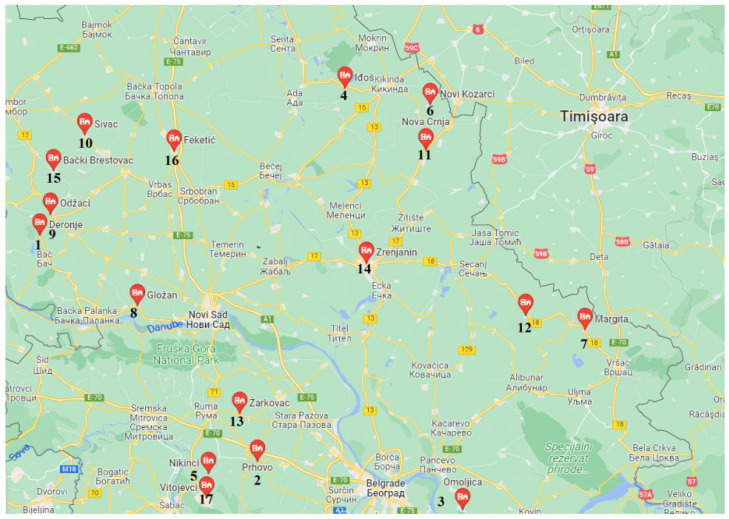
Position of the hunting ground included in the study: 1—Deronje; 2—Prhovo; 3—Omoljica; 4—Iđoš Sajan; 5—Nikinci; 6—Novi Kozarci; 7—Margita; 8—Gložan; 9—Odžaci; 10—Sivac; 11—Nova Crnja; 12—Hajdučica; 13—Žarkovci; 14—Zrenjanin; 15—Bački Brestovac; 16—Feketić; 17—Vitojevci.

**Table 1 animals-12-01249-t001:** The occurrence of lead in samples of hare liver from 17 different locations in the region of Vojvodina (µg/kg of fresh weight), ppb.

	Number		Mean ± Std.Dev		∑	Range	
Location	Y	O	Y	O		Y	O
1	7	2	547 ± 122	569 ± 158	551 ± 120 ^abc^	391–675	457–681
2	2	8	501 ± 38	462 ± 128	470 ± 115 ^ab^	474–528	256–619
3	6	3	869 ± 466	881 ± 600	873 ± 475 ^bcde^	402–1572	211–1367
4	9	3	289 ± 228	472 ± 338	335 ± 256 ^a^	83–740	152–826
5	12	10	707 ± 457	596 ± 393	657 ± 423 ^abcde^	104–1559	234–1503
6	7	11	1535 ± 424	1798 ± 890	1696 ± 740 ^f^	697–2018	475–3700
7	6	0	1803 ± 939	-	1803 ± 939 ^f^	1001–3248	-
8	3	4	1180 ± 864	1170 ± 588	1174 ± 650 ^e^	651–2178	553–1940
9	5	3	804 ± 264	561 ± 141	713 ± 247 ^abcde^	539–1248	403–677
10	7	3	943 ± 416	361 ± 97	768 ± 443 ^abcde^	449–1521	257–450
11	2	3	1000 ± 331	1128 ± 687	1077 ± 518 ^de^	765–1234	702–1920
12	5	2	932 ± 550	1232 ± 40	1018 ± 473 ^cde^	612–1911	1204–1260
13	18	9	810 ± 574	731 ± 388	784 ± 513 ^abcde^	277–2821	259–1251
14	8	5	830 ± 266	468 ± 332	691 ± 334 ^abcde^	426–1189	110–950
15	8	1	539 ± 403	342	517 ± 383 ^abc^	59–1273	342
16	11	7	884 ± 555	722 ± 279	821 ± 463 ^abcde^	280–2339	342–1156
17	6	0	569 ± 398	-	569 ± 398 ^abcd^	71–1118	-
∑	122	74	838 ± 568	841 ± 641	838 ± 595	59–3248	110–3700

Y—young hares; O—old hares; small letters mark significant different concentrations in different locations or ages.

**Table 2 animals-12-01249-t002:** The occurrence of cadmium in samples of hare liver from 17 different locations in the region of Vojvodina (µg/kg of fresh weight), ppb.

	Number		Mean		∑	Range	
Location	Y	O	Y	O		Y	O
1	7	2	148 ± 166	120 ± 63	141 ± 146 ^a^	19–496	75–165
2	2	8	85 ± 66	104 ± 124	100 ± 112 ^a^	39–131	20–392
3	6	3	69 ± 23	138 ± 123	92 ± 73 ^a^	38–97	26–269
4	9	3	99 ± 115	372 ± 395	167 ± 231 ^ab^	13–385	87–823
5	12	10	184 ± 162	285 ± 143	230 ± 204 ^ab^	74–654	72–899
6	7	11	134 ± 66	204 ± 82	177 ± 82 ^ab^	49–246	94–317
7	6	0	327 ± 211	-	327 ± 211 ^abc^	104–685	-
8	3	4	556 ± 746	297 ± 82	408 ± 456 ^bc^	66–1414	214–408
9	5	3	643 ± 523	393 ± 58	549 ± 417 ^c^	101–1183	344–457
10	7	3	233 ± 196	622 ± 149	349 ± 273 ^abc^	48–645	345–829
11	2	3	508 ± 710	987 ± 231	796 ± 470 ^d^	6–1010	853–1254
12	5	2	191 ± 156	170 ± 1	185 ± 128 ^ab^	2	170–171
13	18	9	203 ± 193	467 ± 402	272 ± 299 ^ab^	4–860	58–1113
14	8	5	239 ± 166	259 ± 139	247 ± 150 ^ab^	103–489	124–1113
15	8	1	110 ± 84	308	185 ± 96 ^a^	0–227	308–666
16	11	7	169 ± 185	232 ± 144	194 ± 167 ^ab^	0–557	67–487
17	6	0	110 ± 150	-	110 ± 150 ^a^	0–383	-
∑	122	74	208 ± 246 ^a^	303 ± 272 ^b^	243 ± 259	0–1414	20–1254

Y—young hares; O—old hares; small letters mark significant different concentrations in different locations or ages.

**Table 3 animals-12-01249-t003:** Two-way ANOVA showing variation in heavy metals between the location and age in the samples of hare livers.

	Source	DF	F	*P*
Pb	Location	16	7.6168	<0.005
	Age	1	0.7618	0.3803
	Location	16	4.1907	<0.005
Cd	Age	1	7.8438	<0.005

## Data Availability

Not applicable.

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
