# Peer review of "The European Hare (Lepus europaeus) as a Biomonitor of Lead (Pb) and Cadmium (Cd) Occurrence in the Agro Biotope of Vojvodina, Serbia"

_animals, 2022, doi:10.3390/ani12101249_

Round 1
Reviewer 1 Report
Comments to the Authors of manuscript number: animals-1694529 entitled “The European hare (Lepus europaeus) as biomonitor of lead (Pb) and cadmium (Cd) occurrence in the agro biotope of Vojvodina, Serbia”.
The authors have presented a study on wild animals, hares which in Authors opinion are exposed to heavy metals pollution due to the fact that they drink water and eat food which are uncontrolled for the presence of Pb or Cd . But there should be written about hunting, which is the main source of Pb presence in nature. Shot with a bundle of pellets to birds - these are several hundred small pieces of lead in the wild forest and intoxication for all wild animals. Animals that come to hunting area will eat deadly poisonous lead pellets. Cadmium appears in solutions of soils fertilized with municipal sewage and in soils of industrial regions. The biological accumulation of cadmium in aquatic vegetation is particularly intense.
- L 96 – there should be exactly written what heavy metal
- L 102 – it is not goel, it should be removed and presented in material and methods
- L 161 – using any abbreviation each should be explained. Authors have to present what criteria were taken under the consideration and in to their relation present obtained results
- Tables – data have to be presented traditionally not only as mean value but e.g. mean ± SD or SEM. The range can be stay
- L 201 – rephrase the sentence do not start with they
- L 218 – I think that industrial development has been finished many years ego. The most serious cadmium contamination, often of a local nature (up to 40 km), is associated with the non-ferrous metal industry, especially zinc, lead and copper. Are these in Poland?
- Hunting in general has important role nowadays in heavy metals pollution in wild nature.
- L 220- what is this?
- L 223-224- exactly what region and the reference should be added
- L 233-235 – the role of the use of pellets with Pb during hunting should be presented.
- L 236-237 – Authors must present the statistic related to the hunting in these places
- L 245 – the reference should be added
- L 245-247 – really? For this reason young animals eat more elements from environment and deposit heave metals in liver?
- L 247 -248 what statistical difference of the age. Animals were divided only into young and older. This sentence is fouls, it should be omitted
- The content of cadmium in individual tissues is a function of its concentration in food, and in the short term it accumulates mainly in the kidneys. After the body takes up cadmium for a long time, it accumulates in the liver and then in the bones. Animal organisms absorb 0.5-5.3% of cadmium supplied with food. Why kidneys were not investigated?
- The degree of cadmium contamination by dust emitted from the combined heat and power plant is directly related to its content in coals. Chemical fertilizers and their production constitute a special position in soil contamination with cadmium. What is their location in relation to these 17 places investigated?
- where is the statistical analysis for age related differences? In tables is only for the sum of the Pb or Cd occurrence.
- L 269 this conclusion is not true
Author Response
Dear Editor and Reviewers,
Authors wish to thank the reviewers for their comments and recommendations concerning our manuscript titled “The European hare (Lepus europaeus) as biomonitor of lead (Pb) and cadmium (Cd) occurrence in the agro biotope of Vojvodina, Serbia”.
The comments are all valuable and very helpful for revising and improving our paper, as well as the important guiding significance to our research.
Please find below the list of changes we have made according to the suggestions of the reviewers.
- Line 96
Comment
There should be exactly written what heavy metal
We added two sentences:
- In more than half samples of spinach the concentration of cadmium exceeded the maximum permissible concentration. Also, in 46% samples of same vegetable, the lead concentration also exceeded the maximum permissible concentration.
- Line 102
L 102 – it is not goal, it should be removed and presented in material and methods
- Explanation
- The age of the hares is very important for this research and we believe that the procedure of age determination should be part of this section.
- Line 161
161 – using any abbreviation each should be explained. Authors have to present what criteria were taken under the consideration and into their relation present obtained results
Abbreviation which was used in this line has already been explained in Materials and methods section (line 147)
- Tables
data have to be presented traditionally not only as mean value but e.g. mean ± SD or SEM. The range can be stay
We corrected as suggested.
- L201
L 201 – rephrase the sentence do not start with they
We rephrased the sentence that now reads:
A maximum value of 1.72 mg/kg fw was found in samples from the location Sombor which is in Bačka, Vojvodina.
- L 218 – I think that industrial development has been finished many years ego. The most serious cadmium contamination, often of a local nature (up to 40 km), is associated with thenon-ferrous metal industry, especially zinc, lead and copper. Are these in Poland
Study was done in Poland and they concluded that increased level of cadmium and lead are result of industrial development.
The sentence Researchers from Poland, in the industrially developed region found lead in samples of hare’s liver in the range 0.30-3.63 mg/kg has been changed to: Researchers from Poland, found lead in samples of hare’s liver in the range 0.30-3.63 mg/kg. The highest amount of lead was found in the livers of hares taken north and northwest from the Cracow, probably due to the vicinity of heavy industry. The average lead content was 1.24 ± 0.59 mg/kg.
- Hunting in general has important role nowadays in heavy metals pollution in wild nature
We added several sentences discussing this point in the introduction and discussion
Use of lead-ammunition in hunting is also considered to be source of the environmental pollution. Kanstrup et al., (2018) (45) claim that lead ammunition has adverse effects on wildlife, their habitat or human health, and for that reason is issue for sustainable hunting and threat to conservation efforts. Use of such ammunition is considered to be source of higher lead concentration in hunted animals (46, 47, (48). Although samples in this study consist only from undamaged livers taken immediately after the hunts and only some hunters used lead ammunition, there is a possibility that pellet fragments contaminated the specimens which affected the obtained findings
- L 220 what is this
We believe that the reviewer was referring to Western Zachodni location of the research mentioned in the discussion. We changed the sentence that now reads:
Moreover, the average concentration in one of the four investigated regions (location Western Zachodni) was 2.27 ± 1.59 mg/kg ww, with the maximum found level of 6.63 mg/kg ww (34). L 223-224
- L 233-235 – the role of the use of pellets with Pb during hunting should be presented
We added several sentences discussing this point in the introduction and discussion
Use of lead-ammunition in hunting is also considered to be source of the environmental pollution. Kanstrup et al., (2018) (45) claim that lead ammunition has adverse effects on wildlife, their habitat or human health, and for that reason is issue for sustainable hunting and threat to conservation efforts. Use of such ammunition is considered to be source of higher lead concentration in hunted animals (46, 47, (48). Although samples in this study consist only from undamaged livers taken immediately after the hunts and only some hunters used lead ammunition, there is a possibility that pellet fragments contaminated the specimens which affected the obtained findings
- L 236-237 – Authors must present the statistic related to the hunting in these places
Statistics regarding the age and location are shown in the table 3. The P value is significant for location for lead and cadmium
The sentence This is the most probable reason for the statistically significant differences in the average concentration of lead and cadmium regarding location has been changed to This is the most probable reason for the statistically significant differences (P<0.05) showed in the table 3 in the average concentration of lead and cadmium regarding location.
- L 245 – the reference should be added
The reference is added on the end of sentence
The researchers concluded that younger animals have increased need for minerals, especially calcium whose metabolic pathways are closely related with toxicokinetics of lead (38).
- L 245-247 – really? For this reason, young animals eat more elements from environment and deposit heave metals in liver?
This is sentence from L 243-247. Researchers found the explanation in better absorption of lead in younger animals (33). It is a similar situation with other groups of wild animals; younger animals had higher levels of lead in liver (wild boar and roe deer). The researchers concluded that younger animals have increased need for minerals, especially calcium whose metabolic pathways are closely related with toxico kinetic of lead.
It is explained that due to the increase need for calcium which is very similar with the metabolic of lead absorption is higher not that animals eat more elements
- L 247 -248 what statistical difference of the age. Animals were divided only into young and older. This sentence is fouls, it should be omitted
Life span of hare is relatively short, and it is not first time that age of hares is divided only in two groups. Massányi et al, 2015 (cited in our paper), also divided the age of hares only in two groups and statistically analyzed the effect of age. The manuscript has 51 citations in Scopus.
- The content of cadmium in individual tissues is a function of its concentration in food, and in the short terms it accumulates mainly in the kidneys. After the body takes up cadmium for along time, it accumulates in the liver and then in the bones. Animal organisms absorb 0.5-5.3% of cadmium supplied with food. Why kidneys were not investigated?
Unfortunately, during the hunting some of the samples of kidneys were damaged by the shots so the samples of kidneys were excluded from the research.
- The degree of cadmium contamination by dust emitted from the combined heat and power plant is directly related to its content in coals. Chemical fertilizers and their production constitute a special position in soil contamination with cadmium. What is their location in relation to these 17 places investigated
The purpose of this research was to investigate occurrence of cadmium and lead depending on the location and age, to see is there any differences. In some further research we will take into consideration source of contamination on the locations
- where is the statistical analysis for age related differences? In tables is only for the sum of the Pb or Cd occurrence.
In the table 3 it is Two-way ANOVA showing variation in heavy metals between the location and age in the samples of liver hares. Furthermore, we indicated differences between the ages in table 2 for cadmium.
- L 269 this conclusion is not true
There was a statistically significant difference among sample locations, while 268 results between different ages of hares appeared not to be significantly different
In the table of 3 it is shown differences depending on the location and age. The age was not statistically significant for lead occurrence in hare’s liver.
Once again, thank you very much for your comments and suggestions.
We look forward to hearing from you at your earliest convenience.
Yours sincerely,
Dr. Miroslava Polovinski Horvatović
Reviewer 2 Report
Comments on the manuscript entitled “The European hare (Lepus europaeus) as biomonitor of lead (Pb) and cadmium (Cd) occurrence in the agro biotope of Vojvodina, Serbia”, by Beuković et al.
This paper on the effect of pollution on wildlife is very interesting. Nevertheless, there are occasional points that need to be addressed to improve the comprehensibility of this manuscript.
Minor changes
Line 74 - Please amend “Lepus europaeus” to “Lepus europaeus”
Lines 100-104 - Please fuse the first and the last sentences “For this research a total of 196 European hare’s liver samples were collected from locations in Vojvodina, during the regular hunting season.“
Line 132- Remove “by” from the sentence.
Line 135 and lines 146-148 - Please use the verbs in the past simple.
Line 144 - Replace “Cadmium working solutions were ranged from 0.2 to 2.0” to “Cadmium working solutions ranged from 0.2 to 2.0”
Line 214 - Please amend “Lepus europaeus” to “Lepus europaeus”
Line 233 - Please clarify: “The range of hare is relatively small.”
Lines 242 - Please amend these sentences to increase comprehensibility: “Sometimes higher concentration of lead in edible tissues can be found in the younger animals. Researchers found the explanation in better absorption of lead in younger animals (33). It is a similar situation with other groups of wild animals; younger animals had higher levels of lead in liver (wild boar and roe deer).”
Author Response
Dear,
We are sending correction
Line 74-corrected
Lines 100-104-Corrected
Line 132-Correctcd
Line 135 and lines 146-148-Corrected
Line 144-Corrected
Line 233-Corrected, changed from The range of hare is relatively small to Compared with the other wild animals as swine or deer the range of hare is relatively small.
In this sentence the main meaning is that compared to the other wild animals due to the range hare is good for the assessment for the occurrence of heavy metals
Line 242
We changed Researchers found the explanation in better absorption of lead in younger animals to Researchers found the explanation in higher absorption of lead in younger animals.
Researchers dealing with this topic explained that due to very similar kinetics of lead and calcium, in younger animals due to the higher absorption of lead can be higher concentration in liver compared with the older animals
Round 2
Reviewer 1 Report
The manuscript has been improved significantly.